# Twelve-month rehospitalization after IBD-related hospitalization in Germany: A retrospective cohort study using administrative hospital data

Christian Weigel[1‡*], Sven H. Loosen[1‡*], Kaneschka Yaqubi[1], Petra May[1], Tom Luedde[1], Christoph Roderburg[1], Karel Kostev[2]

**1** Department of Gastroenterology, Hepatology and Infectious Diseases, University Hospital Duesseldorf, Medical Faculty of Heinrich Heine University Duesseldorf, Duesseldorf, Germany, **2** Epidemiology, IQVIA, Frankfurt, Germany

‡ These authors share first authorship on this work.
* christian.weigel@med.uni-duesseldorf.de (CW); Sven.Loosen@med.uni-duesseldorf.de (SHL)

## Abstract

### Background/Aims

Rehospitalization is an important indicator of disease burden and healthcare utilization in inflammatory bowel disease (IBD), yet real-world data from Germany remain scarce. This study aimed to describe 12-month rehospitalization rates after IBD-related inpatient treatment in Germany and to identify administrative and clinical factors recorded in hospital ICD-10 coding that are associated with rehospitalization.

### Methods

In this multicenter retrospective cohort study, anonymized administrative data from 49 German hospitals (Section 21 dataset, 2019–2023) were analyzed. Patients hospitalized with a primary diagnosis of Crohn's disease (ICD-10 K50) or ulcerative colitis (ICD-10 K51) were included. The primary endpoint was any rehospitalization within 12 months after the index admission. Univariable and multivariable logistic regression models were used to determine factors of rehospitalization.

### Results

A total of 3,202 patients were included (51.3% Crohn's disease, 48.7% ulcerative colitis; mean age 44.2 years; 50.1% female). The 12-month rehospitalization rate was 22.6%, with a median time of 92 days (IQR 61–184). Independent factors in adjusted analyses were admission to a surgical department (AOR 1.79), peritoneal adhesions (AOR 1.98), type 2 diabetes mellitus (AOR 1.54), and hypokalemia (AOR 1.30). Age, sex, and IBD subtype showed no association.

**Data availability statement:** The data underlying this study are not publicly available due to legal and contractual restrictions. The dataset is owned by IQVIA (Frankfurt, Germany) and contains anonymized hospital administrative data derived from the German Section 21 dataset. Under the data use agreement between IQVIA and the contributing hospitals, the data cannot be shared publicly or deposited in open repositories. Researchers who meet the criteria for access to confidential data may submit requests to IQVIA. Data access requests should be directed to the IQVIA Data Access/ Compliance Office (IQVIA, Frankfurt, Germany, Mail: info.germany@iqvia.com). Access is subject to approval by IQVIA and may require a data use agreement and associated fees.

**Funding:** The author(s) received no specific funding for this work.

**Competing interests:** The authors have declared that no competing interests exist.

## Conclusion

Using administrative hospital data, we identified several ICD-10–coded comorbidities and treatment characteristics associated with higher rehospitalization rates. These findings highlight subgroups that may benefit from improved post-discharge management, although clinical interpretation is limited by the administrative nature of the dataset.

## Introduction

Inflammatory bowel disease (IBD), encompassing Crohn's disease (CD) and ulcerative colitis (UC), is a chronic inflammatory disorder of the gastrointestinal tract associated with substantial morbidity and healthcare utilization [1]. Estimates of the global IBD prevalence range from 4.9 to 6.8 million individuals, reflecting differences across datasets, with the largest patient populations observed in China and the United States [2,3]. The prevalence of IBD continues to rise and is projected to approach 1% of the population in several industrialized regions, such as Europe and North America, within the coming decade [4]. Annual direct healthcare costs for IBD, driven by outpatient care, hospitalizations, and pharmacotherapy, are considerable in high-income countries, averaging between $9,000 and $12,000 per patient [5].Optimal management of CD and UC focuses on inducing and maintaining remission, which may in turn reduce the need for medical interventions and overall healthcare costs [6].

Rehospitalization following an initial IBD-related admission is increasingly recognized as a critical indicator of disease course, quality of care, and healthcare utilization. High rates of rehospitalization not only reflect ongoing disease activity but also contribute substantially to healthcare costs and resource allocation [7]. Internationally, hospitalizations affect a significant proportion of IBD patients and account for a disproportionate share of healthcare expenditures [8]. While most patients experience only occasional hospitalizations, a subset of high-need patients undergo frequent admissions, many of which may be preventable, highlighting the burden of recurrent hospitalizations [9]. Although rehospitalization patterns in IBD have been described in individual international studies, comparable data from Germany are limited. Accordingly, this multicenter retrospective cohort study aimed to describe 12-month rehospitalization rates after IBD-related inpatient treatment in Germany and to identify administrative and clinical factors recorded in hospital ICD-10 coding that are associated with rehospitalization.

## Methods

### Data source

This multicenter retrospective cohort study utilized data from the IQVIA hospital database, which contains Section 21 datasets from 49 hospitals across Germany. These hospitals include specialized, primary care, maximum care, standard care, and university institutions. The Section 21 dataset comprises patient information submitted by hospitals to the Institute for the Hospital Remuneration System (InEK)

in a standardized format, in accordance with Section 21 of the German Hospital Compensation Act (KHEntgG). To prevent any possibility of identifying individual patients through searches in the InEK data browser, the case data were restricted to the information essential for the analysis (such as limited demographic variables, diagnoses, and procedures) and were further anonymized. For example, age was provided only in predefined categories rather than as exact values. In addition, categories or subgroups with fewer than four data entries were not displayed in order to maintain data privacy. As only anonymized data were used, informed consent from individual patients or their legal guardians was not required.

## Study population

We conducted a multicenter retrospective cohort study including all patients with a primary diagnosis of inflammatory bowel disease (IBD), defined as Crohn's disease (ICD-10: K50) or ulcerative colitis (ICD-10: K51), who were hospitalized between January 2019 and December 2023 in one of the participating hospitals. For individuals with multiple admissions during the study period, only the first hospitalization was considered the index event. Patients were followed for up to 12 months after discharge to determine whether a rehospitalization occurred.

Because the Section 21 dataset does not provide individual-level follow-up information beyond subsequent inpatient encounters, patients without a recorded rehospitalization could not be differentiated with certainty regarding whether they remained event-free, were hospitalized elsewhere, or died after discharge. For patients discharged in 2023, follow-up was truncated at the end of the available dataset (December 2023). These patients were therefore included in the analysis, but their follow-up time may have been shorter than 12 months. This limitation is inherent to the structure of the administrative dataset and is acknowledged in the interpretation of the results.

## Study outcome

The primary outcome was any rehospitalization within 12 months after the index hospitalization. Rehospitalization was defined as any subsequent inpatient admission recorded in the dataset, irrespective of diagnosis. Rehospitalization rates were calculated overall and stratified by age group, sex, and IBD subtype. Time to rehospitalization was assessed descriptively.

## Statistical analyses

Descriptive statistics were used to summarize baseline characteristics. Continuous variables are presented as means with standard deviations or medians with interquartile ranges, depending on distribution. Categorical variables are reported as absolute numbers and percentages. All comorbidities were identified exclusively through ICD-10-GM codes documented in the electronic hospital records, including iron deficiency anaemia (D50), gastritis/duodenitis (K29), intestinal obstruction (K56), gastrointestinal hemorrhage (K92.2), peritoneal adhesions (K66.0), type 2 diabetes mellitus (E11), hypothyroidism (E03), volume depletion (E86), hypokalemia (E87.6), and hypertension (I10). The Section 21 dataset does not contain laboratory values, diagnostic thresholds, or information on how these diagnoses were clinically established. Associations between demographic or clinical characteristics and the likelihood of rehospitalization were examined using univariable and multivariable logistic regression models. Odds ratios (ORs) with 95% confidence intervals (CIs) and corresponding p-values were calculated for each predictor. P-values for the ORs were derived from Wald $\chi^2$ tests. All variables with a prevalence of at least 5% were considered for inclusion in the multivariable model. Multicollinearity was assessed using variance inflation factors (VIFs), all of which were <1, indicating no relevant collinearity.

Because patient identifiers are hospital-specific and the dataset is provided in aggregated form, clustering by hospital could not be implemented. However, all hospitals contribute data under standardized national coding and reimbursement regulations, reducing the likelihood of systematic inter-hospital variability that would bias regression estimates.

All statistical tests were two-sided, and a p-value <0.05 was considered statistically significant. Analyses were performed using SAS version 9.4 (SAS Institute, Cary, NC, USA).

## Ethics declarations

The database used includes only anonymized data in compliance with applicable data protection laws. German law allows the use of anonymous electronic medical records for research purposes. According to this legislation, it is not necessary to obtain informed consent from patients or approval from a medical ethics committee for this type of observational study that contains no directly identifiable data. Because patients were only queried as aggregates and no protected health information was available for queries, no Institutional Review Board approval was required for the use of this database or the completion of this study.

## Results

### Baseline characteristics

A total of 3,202 patients were included (1,642 with Crohn's disease and 1,560 with ulcerative colitis, Fig 1). The mean age was 44.2 years (SD 21.4), and 50.1% of patients were female. The median length of hospital stay was 5 days. Overall, 47.0% of patients were treated in internal medicine, 19.3% in gastroenterology, 10.0% in pediatrics, 21.7% in surgery, and 1.9% in other departments. The most common comorbidities were hypertension (20.4%), gastritis/duodenitis (18.0%), and hypokalemia (14.9%, Table 1).

### Rehospitalization rates

The overall 12-month rehospitalization rate was 22.6%. Rates increased from 15.9% in patients younger than 18 years to 30.4% among those older than 60 years. Rates were similar between women (23.1%) and men (22.2%). The median time to rehospitalization was 92 days (IQR 61–184 days). Among the rehospitalized patients, 44.5% were admitted again due to IBD as the primary diagnosis, 14.1% due to other gastrointestinal diseases, and 5.7% due to cardiovascular diseases.

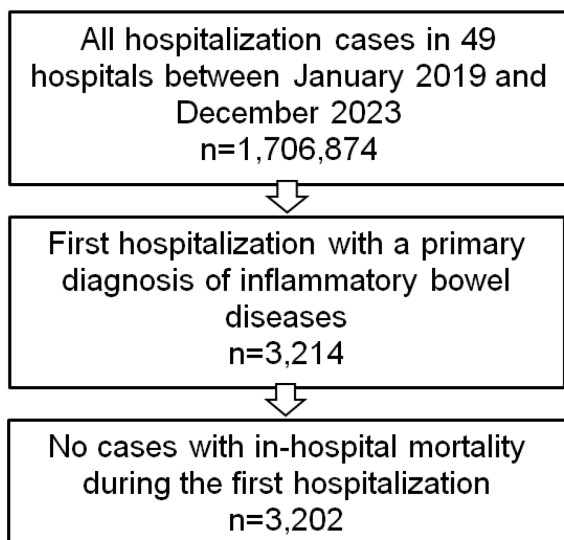

**Fig 1. Selection of study patients.** Selection of patients hospitalized between January 2019 and December 2023 in 49 German hospitals. From 1,706,874 hospitalizations, 3,214 patients with a primary diagnosis of inflammatory bowel disease (ICD-10: K50 or K51) were identified. After exclusion of patients with in-hospital mortality during the index admission, 3,202 patients were included in the final analysis.

**Table 1. Baseline characteristics of the study sample.**

| Variable | Hospitalized patients (N = 3,202) |
|---|---|
| Mean age (standard deviation) | 44.2 (21.4) |
| <18 years | 371 (11.6) |
| 18-30 years | 634 (19.8) |
| 31-40 years | 515 (16.1) |
| 41-50 years | 398 (12.4) |
| 51-60 years | 491 (15.3) |
| >60 years | 793 (24.8) |
| Female | 1604 (50.1) |
| Male | 1598 (49.9) |
| Hospital department | |
| Internal medicine | 1505 (47.0) |
| Gastroenterology | 619 (19.3) |
| Pediatric | 321 (10.0) |
| Surgery | 696 (21.7) |
| Other departments | 61 (1.9) |
| Length of hospital stay in days (median, IQR) | 5 (2-9) |
| Main diagnosis | |
| Crohn's disease | 1642 (51.3) |
| Ulcerative colitis | 1,560 (48.7) |
| Secondary diagnoses | |
| Iron deficiency anaemia | 268 (8.4) |
| Gastritis and duodenitis | 575 (18.0) |
| Intestinal obstruction | 241 (7.5) |
| Gastrointestinal hemorrhage | 261 (8.2) |
| Peritoneal adhesions | 209 (6.5) |
| Type 2 diabetes mellitus | 212 (6.6) |
| Hypothyroidism | 222 (6.9) |
| Volume depletion | 310 (9.7) |
| Hypokalemia | 477 (14.9) |
| Hypertension | 653 (20.4) |

Data are absolute numbers (percentages) unless otherwise specified.

All other disease groups (including psychiatric, neurological, endocrinological, and others) each accounted for less than 5% of primary diagnoses.

### Factors associated with increased rehospitalization rates

In univariable analysis, patients aged 51–60 years (OR 2.03, 95% CI 1.44–2.85) and those over 60 years (OR 2.31, 95% CI 1.68–3.17) had higher odds of rehospitalization compared with those younger than 18 years. Surgical department admission was associated with increased odds (OR 1.84, 95% CI 1.50–2.26), while treatment in pediatrics was associated with lower odds (OR 0.68, 95% CI 0.48–0.94) compared with internal medicine. Among comorbidities, peritoneal adhesions (OR 2.88, 95% CI 2.16–3.83), type 2 diabetes mellitus (OR 2.07, 95% CI 1.54–2.78), hypokalemia (OR 1.60, 95% CI 1.29–2.00), and hypertension (OR 1.55, 95% CI 1.28–1.88) were associated with greater odds of rehospitalization.

In the multivariable model, several clinical characteristics demonstrated significant associations with rehospitalization within 12 months. Admission to a surgical department was associated with higher odds of subsequent hospitalization (adjusted OR (AOR) 1.79, 95% CI 1.39–2.30). The presence of peritoneal adhesions also showed a notable association (AOR 1.98, 95% CI 1.40–2.81), as did type 2 diabetes mellitus (AOR 1.54, 95% CI 1.12–2.13) and hypokalemia (AOR 1.30, 95% CI 1.04–1.64). No statistically significant associations were observed for patient age, sex, IBD subtype, or other comorbidities after adjustment.

## Discussion

The 12-months rehospitalization rate in this multicenter German cohort is comparable to findings from other Western healthcare systems, although reported rates vary considerably depending on study design and follow-up duration. In a large meta-analysis conducted in the United States and Australia, the 30-day readmission rate was 18.1%, while the 90-day rate was notably higher at 26.0% [10]. The observed increase in the readmission rate within the 1- to 3-month interval is not unexpected and is consistent with the findings of the present study, which demonstrated a median time to rehospitalization of 92 days. A Canadian population-based study reported an annual readmission rate of approximately 20% among patients with IBD, which is comparable to the findings of the present study, where the 12-months rehospitalization rate was 22.6% [11]. In contrast, another North American cohort reported a substantially higher rate of unplanned readmissions, with 39.2% of patients rehospitalized within one year [12]. These findings highlight the heterogeneity in reported rehospitalization rates and show the need for region-specific analyses to support individual targeted healthcare strategies.

With regard to potential factors associated with rehospitalization in patients with IBD, our statistical analyses using univariable and multivariable logistic regression identified several variables of interest, some of which are consistent with findings from previous studies (Table 2). Surgical treatment emerged as one of the strongest factors associated with an increased 12-months rehospitalization. Admission to a surgical department during the index hospitalization was associated with rehospitalization, suggesting that patients with complicated disease courses represent a clinically vulnerable subgroup. In line with our results, other studies have similarly reported an increased risk of rehospitalization in patients with fistulating or perianal disease, the need for surgery, ileostomy in CD, colostomy in UC, and large intestinal resection in general [13–16].

The association between surgical treatment and rehospitalization likely reflects the burden of advanced or complicated disease, which often entails recurrent clinical instability. In this context, electrolyte disturbances are common sequelae in IBD, particularly in patients with high intestinal output, active colitis, infectious complications, or short bowel syndrome following resection. Consistent with these mechanisms, hypokalemia was independently associated with an increased rehospitalization rate in in both univariable and multivariable regression analyses. This finding underscores the clinical importance of early detection and correction of electrolyte imbalances as part of post-discharge management in IBD.

Age showed an association with rehospitalization rates in univariable analysis, with patients aged ≥51 years demonstrating higher overall rehospitalization rates. However, this effect was not sustained after adjustment for comorbidities. These findings suggest that age alone is not an independent factor associated with rehospitalization in IBD. This observation is consistent with previous studies reporting inconsistent age effects, with some cohorts indicating lower rehospitalization risk in older patients, while others identified younger age as a risk factor in CD [13,14]. Taken together, these findings indicate that age alone may be less relevant than multimorbidity and overall disease complexity in explaining increased rehospitalization rates, highlighting the need for stratification models that incorporate comorbidity indices rather than patients' age.

Nutritional and metabolic status have gained increasing attention as determinants of outcomes in IBD. Alterations in body composition, on the one hand muscle wasting, on the other hand excess obesity, are frequently observed in patients with malnutrition and sarcopenia and are associated with impaired quality of life [17]. Furthermore, both underweight/

**Table 2. Association between demographic and clinical variables and 12-months rehospitalization in patients hospitalized for IBD (multivariable logistic regression).**

| Variable | Re-hospitalization rate (%) | Crude OR (95% CI) | p-value | Adjusted OR (95% CI) | p-value |
|---|---|---|---|---|---|
| <18 years | 15.9 | Reference | | Reference | |
| 18-30 years | 16.6 | 1.05 (0.74-1.49) | 0.785 | 0.69 (0.33-1.45) | 0.326 |
| 31-40 years | 20.4 | 1.35 (0.95-1.92) | 0.091 | 0.83 (0.39-1.76) | 0.629 |
| 41-50 years | 19.6 | 1.29 (0.89-1.87) | 0.182 | 0.76 (0.36-1.63) | 0.486 |
| 51-60 years | 27.7 | 2.03 (1.44-2.85) | <0.001 | 1.21 (0.57-2.54) | 0.625 |
| >60 years | 30.4 | 2.31 (1.68-3.17) | <0.001 | 1.33 (0.63-2.83) | 0.455 |
| Female | 23.1 | Reference | | Reference | |
| Male | 22.2 | 0.95 (0.80-1.12) | 0.536 | 1.00 (0.84-1.90) | 0.975 |
| Internal medicine | 20.3 | Reference | | Reference | |
| Gastroenterology | 22.0 | 1.11 (0.88-1.39) | 0.379 | 1.17 (0.92-1.47) | 0.196 |
| Paediatry | 14.6 | 0.68 (0.48-0.94) | 0.021 | 0.77 (0.35-1.69) | 0.510 |
| Surgery | 31.9 | 1.84 (1.50-2.26) | <0.001 | 1.79 (1.39-2.30) | <0.001 |
| Crohn's disease | 21.4 | Reference | | Reference | |
| Ulcerative colitis | 23.9 | 1.16 (0.98-1.36) | 0.087 | 1.14 (0.96-1.37) | 0.142 |
| Iron deficiency anaemia | 20.5 | 0.87 (0.64-1.19) | 0.394 | 1.04 (0.76-1.44) | 0.796 |
| Gastritis and duodenitis | 22.1 | 0.96 (0.78-1.20) | 0.742 | 1.09 (0.87-1.37) | 0.456 |
| Intestinal obstruction | 25.7 | 1.20 (0.89-1.63) | 0.230 | 0.85 (0.61-1.20) | 0.356 |
| Gastrointestinal hemorrhage | 22.6 | 1.00 (0.74-1.35) | 0.999 | 1.03 (0.75-1.42) | 0.853 |
| Peritoneal adhesions | 43.5 | 2.88 (2.16-3.83) | <0.001 | 1.98 (1.40-2.81) | <0.001 |
| Type 2 diabetes mellitus | 36.3 | 2.07 (1.54-2.78) | <0.001 | 1.54 (1.12-2.13) | 0.008 |
| Hypothyroidism | 27.9 | 1.36 (1.00-1.84) | 0.051 | 1.08 (0.78-1.49) | 0.651 |
| Volume depletion | 24.2 | 1.10 (0.84-1.45) | 0.484 | 1.05 (0.79-1.40) | 0.732 |
| Hypokalemia | 30.2 | 1.60 (1.29-2.00) | <0.001 | 1.30 (1.04-1.64) | 0.024 |
| Hypertension | 29.1 | 1.55 (1.28-1.88) | <0.001 | 1.04 (0.82-1.32) | 0.735 |

sarcopenia and obesity, have been identified in US studies as being associated with increased rehospitalization rates, poorer clinical outcomes, and higher annual healthcare costs [14,16,18,19]. Although anthropometric and body composition data were not available in our dataset, metabolic comorbidities such as type 2 diabetes and hypertension were associated with an increased rehospitalization rate in adjusted analyses. These findings indirectly support the hypothesis that metabolic dysfunction may contribute to recurrent hospitalization among IBD patients.

Neither sex nor the IBD subtype was independently associated with an increased rehospitalization rate in our cohort and evidence from the literature on these factors is inconsistent. Only few studies have reported an impact of sex on the risk of rehospitalization. For example, one study found that male patients had a higher risk of readmission within 30 days after discharge. The same study also reported an increased risk for patients with CD within the same observation period [13]. Consistent with most previous studies, our analysis showed that neither sex nor the IBD subtype is associated with higher rehospitalization rates, suggesting that rehospitalization is more driven by comorbidities and disease complications.

This study has several limitations that should be acknowledged. First, although the Section 21 dataset includes an anonymous patient identifier, this identifier is hospital-specific and cannot be linked across institutions. Consequently, patients without a recorded rehospitalization in the participating hospital could not be distinguished from those who may have been hospitalized in non-participating hospitals or who died after discharge. For patients discharged in 2023, follow-up was truncated at the end of the observation window, resulting in potentially shorter observation periods for this

subgroup. Second, although the dataset covers 49 hospitals across different levels of care, clustering by hospital could not be implemented because the data were provided in aggregated form. Nevertheless, all hospitals submit data under standardized national coding and reimbursement regulations, reducing the likelihood of systematic inter-hospital variability that would materially bias regression estimates. Third, important clinical variables such as disease activity, medication use, surgical history, nutritional status, and laboratory parameters were not available in the administrative dataset. This limited the ability to adjust for disease severity or to conduct meaningful subgroup analyses. Given these constraints, additional stratified analyses—for example by IBD subtype—were not performed to avoid over-interpretation of incomplete data. Fourth, although multicollinearity was assessed and excluded (all VIF values <1), residual confounding cannot be fully ruled out in observational research. Finally, the inability to differentiate between planned and unplanned rehospitalizations, as well as the absence of outpatient data, may have influenced the interpretation of rehospitalization patterns.

This study also has notable strengths, including a large sample size, multicenter design across different levels of care, and standardized data acquisition, which increase the robustness and generalizability comorbidities of the findings within the German healthcare system. Adjustment for relevant comorbidities allowed identification of clinically meaningful factors which are associated with higher rehospitalization rates. Importantly, our results highlight the clinical relevance of identifying factors associated with rehospitalization during the index hospitalization.

Using administrative hospital data, we identified several ICD-10–coded comorbidities and treatment characteristics associated with higher rehospitalization rates. These findings highlight subgroups that may benefit from improved post-discharge management, although clinical interpretation is limited by the administrative nature of the dataset.

## Author contributions

**Conceptualization:** Christian Weigel, Karel Kostev.

**Data curation:** Karel Kostev.

**Formal analysis:** Karel Kostev.

**Methodology:** Karel Kostev.

**Supervision:** Tom Luedde, Christoph Roderburg, Karel Kostev.

**Validation:** Kaneschka Yaqubi, Petra May, Tom Luedde.

**Writing – original draft:** Christian Weigel, Sven H. Loosen, Karel Kostev.

**Writing – review & editing:** Christian Weigel, Sven H. Loosen, Kaneschka Yaqubi, Petra May, Tom Luedde, Christoph Roderburg, Karel Kostev.

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
