## [Decision Letter · Decision Letter 0]

6 Feb 2026

Thank you for submitting your manuscript to PLOS ONE. After careful consideration, we feel that it has merit but does not fully meet PLOS ONE’s publication criteria as it currently stands. Therefore, we invite you to submit a revised version of the manuscript that addresses the points raised during the review process.

We look forward to receiving your revised manuscript.

Kind regards,

Marwan Salih Al-Nimer, MD, PhD

Academic Editor

PLOS One

Journal Requirements:

“There was no specific funding for this analysis.”

4. In the online submission form, you indicated that your data is available only on request from a third party. Please note that your Data Availability Statement is currently missing contact details for the third party, such as an email address or a link to where data requests can be made. Please update your statement with the missing information.

5. Please ensure that you refer to Figure 2 in your text as, if accepted, production will need this reference to link the reader to the figure.

6. Please include a separate caption for each figure in your manuscript.

Additional Editor Comments:

1: The title, objective, and conclusion of this descriptive study are inconsistent with the methodology.

2: Complete revision of the methods is necessary. Write in detail the laboratory and clinical diagnosis included in this study, e.g. hypokalemia: how it is diagnosed, whether it is a routine measurement for each visit, what the upper limit is, etc.

3: Revise the statistical analysis section because in the results there is mean, median,, SD........ . What about the p-values for Odds ratios

4: Figure 2 not informative and is poorly presented

5: References are typed inconsistently.

Reviewers' comments:

Reviewer's Responses to Questions

**Comments to the Author**

1. Is the manuscript technically sound, and do the data support the conclusions?

Reviewer #1: Partly

2. Has the statistical analysis been performed appropriately and rigorously?

Reviewer #1: Yes

3. Have the authors made all data underlying the findings in their manuscript fully available?

Reviewer #1: No

4. Is the manuscript presented in an intelligible fashion and written in standard English?

Reviewer #1: Yes

Reviewer #1: Thank you very much for the opportunity to review this manuscript. This study is important and timely, as the findings have practical implications for post-discharge care.

Please consider my suggestions, below:

- in the methods section, the manuscript describes the study design as "cross-sectional" despite including the temporal element of rehospitalizations. The study tracks patients for 12 months to detect rehospitalizations. This constitutes a retrospective cohort study design. I suggest updating the terminology throughout the manuscript.

- Were there incomplete follow-ups, i.e., 12 months beyond December 2023? I suggest adding a line to clarify how were patients admitted in 2023 handled, as well as patients who died after discharge or lost to follow-up.

- Were the 49 hospitals similar in practices and operational environment? I suggest updating the regression model to cluster patients under their hospitals, or add a line of justification as to why clustering is not necessary.

- I suggest adding a line to confirm that VIF was assessed.

- A valuable addition would be considering subgroup analyses, especially that the sample size is large and with a near-equal distribution between Crohn's disease and Ulcerative colitis. (optional)

.

Reviewer #1: No

---

## [Author Response · Author response to Decision Letter 1]

23 Feb 2026

Dear Reviewer,

We sincerely thank you for the careful evaluation of our manuscript entitled “Twelve-month rehospitalization after IBD-related hospitalization in Germany: a retrospective cohort study using administrative hospital data” (PONE-D-25-66720). We appreciate the constructive and insightful comments, which have helped us to improve the clarity, methodological transparency, and overall presentation of our work. Below, we provide a detailed, point-by-point response to all comments raised by you. All changes have been incorporated into the revised manuscript. A marked-up version with track changes and a clean version of the manuscript are submitted separately.

We again thankyou for their thoughtful and constructive feedback. We believe the revisions have substantially strengthened the manuscript. We look forward to your response and the opportunity to contribute to the field of gastroenterology.

With kind regards,

Christian Weigel & Sven H. Loosen

(on behalf of all co-authors)

---

## [Editor Report · Decision Letter 1]

25 Feb 2026

Dear Dr. Christian Weigel,

Thank you for submitting your manuscript to PLOS ONE. After careful consideration, we feel that it has merit but does not fully meet PLOS ONE’s publication criteria as it currently stands. Therefore, we invite you to submit a revised version of the manuscript that addresses the points raised during the review process.

**ACADEMIC EDITOR: Minor revision**

We look forward to receiving your revised manuscript.

Kind regards,

Marwan Salih Al-Nimer, MD, PhD

Academic Editor

PLOS One

Journal Requirements:

Additional Editor Comments (if provided):

Type the references according to the PLoS ONE style.

---

## [Author Response · Author response to Decision Letter 2]

15 Mar 2026

Dear Prof. Al-Nimar, dear Mrs. Maderazo,

Thank you for your message and for the opportunity to revise our manuscript.

Thank you espacially for clarification concerning the data availability statement. The dataset used in this study is owned by IQVIA and was provided under a contractual data use agreement with contributing hospitals. Due to legal and contractual restrictions, including data protection requirements and the proprietary nature of the dataset, the data cannot be shared publicly or deposited in an open repository. We have revised the Data Availability Statement accordingly and now provide a non-author institutional contact point at IQVIA for data access requests.

Kind regards,

Dr. Christian Weigel

---

## [Editor Report · Decision Letter 2]

17 Mar 2026

Twelve‑month rehospitalization after IBD‑related hospitalization in Germany: a retrospective cohort study using administrative hospital data

PONE-D-25-66720R2

Dear Dr. Christian Weigel,

We’re pleased to inform you that your manuscript has been judged scientifically suitable for publication and will be formally accepted for publication once it meets all outstanding technical requirements.

Kind regards,

Marwan Salih Al-Nimer, MD, PhD

Academic Editor

PLOS One
---

## [Editor Report · Acceptance letter]

PONE-D-25-66720R2

PLOS One

Dear Dr. Weigel,

I'm pleased to inform you that your manuscript has been deemed suitable for publication in PLOS One. Congratulations! Your manuscript is now being handed over to our production team.

Kind regards,

on behalf of

Professor Marwan Salih Al-Nimer

Academic Editor

PLOS One